# Associations between Monocyte and T Cell Cytokine Profiles in Autism Spectrum Disorders: Effects of Dysregulated Innate Immune Responses on Adaptive Responses to Recall Antigens in a Subset of ASD Children

**DOI:** 10.3390/ijms20194731

**Published:** 2019-09-24

**Authors:** Harumi Jyonouchi, Lee Geng

**Affiliations:** 1Department of Pediatrics, Saint Peter’s University Hospital (SPUH), New Brunswick, NJ 08901, USA; lgeng@saintpetersuh.com; 2Department of Pediatrics, Rutgers-Robert Wood Johnson medical school, New Brunswick, NJ 08901, USA

**Keywords:** ASD, cytokine, monocyte, β-glucan, T cell cytokine, trained immunity, maternal immune activation

## Abstract

Changes in monocyte cytokine production with toll like receptor (TLR) agonists in subjects with autism spectrum disorders (ASD) were best reflected by the IL-1β/IL-10 ratios in our previous research. The IL-1β/IL-10 based subgrouping (low, normal, and high) of ASD samples revealed marked differences in microRNA expression, and mitochondrial respiration. However, it is unknown whether the IL-1β/IL-10 ratio based subgrouping is associated with changes in T cell cytokine profiles or monocyte cytokine profiles with non-TLR agonists. In ASD (*n* = 152) and non-ASD (*n* = 41) subjects, cytokine production by peripheral blood monocytes (PBMo) with TLR agonists and β-glucan, an inflammasome agonist, and T cell cytokine production by peripheral blood mononuclear cells (PBMCs) with recall antigens (Ags) (food and candida Ags) were concurrently measured. Changes in monocyte cytokine profiles were observed with β-glucan in the IL-1β/IL-10 ratio based ASD subgroups, along with changes in T cell cytokine production and ASD subgroup-specific correlations between T cell and monocyte cytokine production. Non-ASD controls revealed considerably less of such correlations. Altered innate immune responses in a subset of ASD children are not restricted to TLR pathways and correlated with changes in T cell cytokine production. Altered trained immunity may play a role in the above described changes.

## 1. Introduction

Autism spectrum disorder (ASD) is a behaviorally defined syndrome and the effects of genetic and environmental factors that form its pathogenesis, likely vary in individuals with ASD diagnosis. However, recent research is increasingly supporting the role of immune mediated inflammation in its pathogenesis [1,2,3]. In fact, one of the most extensively studied animal models of ASD is generated by inducing sterile inflammation in pregnant rodents through intraperitoneal injection of stimulants of innate immunity, such as endotoxin [4]. This model which is called maternal immune activation (MIA) has been shown to create lasting neurodevelopmental and behavioral changes in offspring, as well as lasting effects on T cell functions [4,5,6]. Perinatal stresses causing immune activation such as maternal infection, during pregnancy, has also been implicated with pathogenesis of ASD and other neuropsychiatric conditions [7,8]. In our clinic, we have also observed that some ASD subjects reveal fluctuating behavioral symptoms and even repeated loss of once acquired cognitive activity following each immune insult. In addition, our previous research indicated an association between fluctuating behavioral symptoms and dysregulated innate immune responses [9,10].

The MIA model relies on the induction of sterile immune activation through non-specific stimulants of innate immunity. However, the lack of antigen (Ag)-specific immune memory in innate immunity, as opposed to adaptive immunity, has made the lasting effects of puzzling. Although neuronal damage caused by innate immune activation in mothers could cause lasting effects in neuronal development, if this is the case, such neurodevelopmental changes will remain static, as seen in patients with cerebral palsy. However, we observed that ASD subjects reveal fluctuating behavioral symptoms following immune insults (mainly microbial infection) repeatedly affect behavioral symptoms along with a high frequency of co-morbid conditions. That is, these ASD subjects develop multi-system inflammation not restricted to the brain (inflammatory subtype) [11]. In addition, direct damage to the brain by MIA would not explain MIA’s lasting effects on T cell functions [6].

The recent discovery of innate immune memory (IIM) may explain many aspects of the lasting effects of MIA. Reprogramming of innate immunity via metabolic and epigenetic changes following initial innate immune stimulus, was first realized as a part of the non-specific effects of vaccinations such as Bacillus Calmette–Guérin (BCG) [12]. Such effects of IIM were termed as trained immunity (TI). It has been shown that TI causes hyper-responsiveness of innate immunity against second stimuli, which can be totally different from the 1^st^ stimuli, since TI is established through metabolic and subsequent epigenetic modulation of innate immune cells [12,13,14,15]. On the other hand, the 1^st^ innate immune stimuli can also cause subsequent hypo-responsiveness of innate immunity (tolerance) as shown in endotoxin tolerance [16]. Innate immune tolerance is also thought to be induced by metabolic and epigenetic regulation [17,18]. Induction of innate immune memory appears to be associated with dose, kinds of stimuli, and other environmental and genetic factors [19].

We have previously reported that some ASD subjects reveal changes in monocyte activation status at the level of cytokine production, expression of microRNA (miRNAs), and mitochondrial respiration [20,21]. Our research has also revealed that the above-described changes are closely associated with production of monocyte cytokines as well as the ratio of IL-1β/IL-10 [20,21]. In these studies, we used a panel of agonists of toll like receptors (TLRs) for stimulating peripheral blood monocytes (PBMo), main innate immune cells in the peripheral blood. It was found that subgrouping of ASD subjects based on IL-1β/IL-10 ratios help identify ASD subjects who have dysregulated innate immune responses; high or low ratio ASD subgroup reveal marked differences in miRNA expression and mitochondrial respiration [20,21]. These findings support the presence of maladapted TI in these ASD children.

In the animal model of TI, with β-glucan derived from *Candida albicans* used as an initial stimulus, the IL-1β family was reported to have a major role in the development of TI [22]. Additionally, β-glucan tends to induce TI, but not tolerance, while endotoxin tends to induce tolerance when given at a low dose [23]. β-glucan is a representative dectin-1 agonist, which stimulates canonical inflammasomes of innate immune cells, leading to cleavage of proinflammatory cytokines, especially IL-1β and IL-18. Inflammasome mediated pathways are different from TLR mediated pathways, but often they exert converging effects on activation of innate immunity [24,25]. It is also noted that miRNAs have important regulatory roles in inflammasome priming and activation [25]. Thus, it can be hypothesized that TI suspected in ASD children will include multiple pathways of innate immunity, including both TLRs and inflammasome mediated pathways.

We also noted that in the low IL-1β/IL-10 ratio ASD subgroup, there is a higher frequency of non-IgE mediated food allergy (NFA) as compared to the normal ratio ASD subgroups [21]. NFA is thought to be associated with cellular immune reactivity to common food proteins, and we have previously reported that ASD children reveal increased T cell cytokine production (TNF-α, IFN-γ, and IL-12) against cow’s milk proteins at a high frequency. Given these findings, it may be hypothesized that exaggerated innate immune responses due to maladapted TI could result in the alternation of adaptive immune responses, resulting in unwanted T cell responses to benign environmental Ags, such as food proteins, and self-Ags.

This study was formulated to examine the above described hypothesis by assessing whether changes in the monocyte cytokine production are detected under β-glucan stimulated cultures in IL-1β/IL-10 based ASD subgroups, and whether T cell cytokine production in response to luminal antigens differ among IL-1β/IL-10 based ASD subgroups. This study determined monocyte cytokine profiles in responses to β glucan, in comparison with their responses to TLR agonists, and T cell cytokine profiles in response to food and candida proteins in ASD (*n* = 152) and control non-ASD (*n* = 41) subjects. Samples from ASD subjects were subdivided into the IL-1β/IL-10 based subgroups (high, normal, and low) as reported previously [20]. The results obtained support our hypotheses described above.

## 2. Results

### 2.1. Clinical Characteristics in the IL-1β/IL-10 Based ASD Subgroups

A summary of clinical characteristics of the ASD subgroups are shown in Table 1. There are no statistical differences in gender frequency and clinical characteristics (ASD severity, cognitive development, sleep, GI symptoms, history of non-IgE mediated food allergy (NFA), seizure disorders, specific antibody deficiency, AR, and asthma) by Chi-Square test (*p* > 0.05) (Table 1). Ages in the ASD subgroups did not differ (*p* > 0.05 by Kruskal–Wallis test). History of NFA tended to be higher in the low IL-1β/IL-10 ratio group than the normal ratio ASD subgroup, but this was not statistically significant (*p* = 0.1286 by Fisher’s exact test).

We also assessed the frequency of prescription medication use in the IL-1β/IL-10 ratio based ASD subgroups. Table 2 summarizes frequency of intake of prescription medications at the time of sample obtainment. For ASD subjects who had samples taken at multiple time points, the medications taken at each time-point was assessed. There was no difference in the frequency of medication intake between the ASD subgroups (*p* > 0.05 by Chi-Square test). Co-variance analysis rejected the effects of the above described clinical characteristics and prescription medication use on IL-1β/IL-10 ratios under all the culture conditions tested in this study.

### 2.2. Changes in Monocyte Cytokine Production in Response to Candida Heat Extract (β-Glucan) the IL-1β/IL-10 Based ASD Subgroups

Since TLR agonists were used to generate IL-1β/IL-10 ratios for subgrouping ASD samples previously, we also assessed the effects of β-glucan, a ligand to dectin 1 which activates innate immunity through inflammasome. We used candida heat extract as a source of β-glucan. One of the reasons that we used β-glucan is that this is widely used for TI induction [12]. Under β-glucan stimulated cultures, significant changes in IL-1β/IL-10 ratios and in the production of IL-1β, IL-10, TNF-α, sTNFRII, and CC-chemokine-ligand-2 (CCL2) were observed (Table 3). There is a tendency of high production of inflammatory cytokines (IL-1β and TNF-α) in the high ratio ASD subgroup, while the low ratio ASD subgroup revealed higher levels of counter-regulatory cytokines (IL-10, sTNFRII, and TGFβ) and CCL2, chemokine under the β-glucan stimulated cultures (Table 3). Therefore, differences were the most apparent between the high vs. low ratio ASD subgroups (Table 3). Such changes were not observed in the production of IL-6, IL-12, IL-23, and CCL7. The changes in monocyte cytokine profiles were more evident with stimulus of β-glucan as compared to monocyte cytokine production without stimuli (Table 3).

We were able to obtain samples at 2–4 different time points in 22 ASD subjects to test longitudinal changes in each subject. With the use of repeated measures of analysis of variance, significant variability was observed in IL-1β/IL-10 ratios under the cultures without stimulus and β-glucan stimulated cultures (F-ratio 5.0 and 3.88, with *p* < 0.01 and *p* < 0.02, respectively).

### 2.3. Changes in T Cell Cytokine Production in the IL-1β/IL-10 Based ASD Subgroups

We then tested whether T cell cytokine production differs among the study groups, including non-ASD controls. Numerically significant differences were found between the study groups in the production of IFN-γ (medium only and in response to candida), TNF-α (in all the culture conditions), and IL-10 (in response to β-LG and gliadin), as shown in Table 4. When the whole ASD samples were compared with non-ASD controls, no significant differences were observed (*p* > 0.05). These changes are mainly due to higher production of TNF-α and IL-10 in the high IL-1β/IL-10 ratio ASD subgroup as compared to other study groups (Table 4). On the other hand, IFN-γ production was lower in the high ratio ASD subgroup, as compared to the normal and low ratio ASD subgroups in the cultures without stimuli or with candida protein (Table 4). In addition, the low ratio subgroup revealed lower IL-10 production in response to gliadin than other study groups (Table 4).

When we assessed the variability of the production T cell cytokines, no significant time-dependent variations were found under all culture conditions, except for IFN-γ production under soy-protein stimulated cultures (F-ratio 4.93, *p* < 0.01).

### 2.4. Associations between T Cell Cytokine Production by Peripheral Blood Mononuclear Cells (PBMCs) and Monocyte Cytokine Production

Since we observed differences in the production of IFN-γ, TNF-α, and IL-10 by peripheral blood mononuclear cells (PBMCs) under cultures stimulated with β-LG, gliadin, and candida, we assessed whether production of these cytokines revealed any correlation with the cytokines produced by PBMo in response to TLR agonists and/or β-glucan.

When IFN-γ production by PBMCs were compared with monocyte cytokine profiles, we observed significant correlations between IFN-γ levels produced by PBMCs and levels of monocyte cytokines produced by PBMo in the ASD subgroups (Table 5). Interestingly, correlations markedly differed among the ASD subgroups, while the non-ASD controls revealed little correlations. Such correlations were most notable between IL-6 and CCL2 production by PBMo and IFN-γ production by PBMCs in response to candida proteins in the normal ratio ASD subgroup (Table 5).

As for TNF-α production, production of TNF-α by PBMCs in response to β-LG and gliadin and that by PBMo revealed a significant correlation with TNF-α production by PBMo in response to TLR agonists and β-glucan (Table 6). Such correlations were most notable in the high ratio ASD subgroup. However, such correlations were also noted in the low ratio and non-ASD subgroups as well, but to a lesser extent (Table 6). On the other hand, few correlations were found in the normal ratio ASD subgroups (Table 6). It should be noted that PBMCs from the normal ratio ASD subgroup produced equivalent amounts of TNF-α, as compared to both the low ratio ASD subgroup and the non-ASD controls (Table 4), making it unlikely that the lack of correlations in the normal ASD subgroup is associated with low TNF-α production. The high and low ratio ASD subgroups revealed positive associations between TNF-α production by PBMCs in response to candida protein, and production of IL-1β, IL-6, IL-10, and sTNFRII by PBMo.

When we examined correlations between IL-10 production by PBMCs and monocyte cytokine production, we found different correlating patterns between the ASD subgroups in response to β-LG and gliadin, while the non-ASD controls revealed little correlations (Table 7). In response to candida protein, only the high ratio ASD subgroup revealed positive correlations between IL-10 production by PBMCs and monocyte cytokine production (Table 7). Non-ASD controls revealed positive correlations between IL-10 production by PBMCs and sTNFRII production by PBMo, and negative correlations between IL-10 production by PBMCs and CCL2 production by PBMo.

We also observed significant correlations between IL-12 production by PBMCs and monocyte cytokine production, although IL-12 production did not differ between IL-1β/IL-10 based ASD subgroups (Table 8). Namely, the low ratio ASD subgroup revealed significant correlations between PBMC IL-12 production and PBMo cytokine production, while the other ASD subgroups and non-ASD controls revealed little correlations between these two parameters.

## 3. Discussion

Changes in IL-1β/IL-10 ratios and monocyte cytokine profiles in the ASD subgroups, which we have reported before with the use of TLR agonists, were also found under the β-glucan stimulated cultures in this study. This finding indicates that the innate immune abnormalities observed in some ASD children involve both TLR and inflammasome mediated pathways. T cell cytokine profiles (IFN-γ, TNF-α, and IL-10) also differed among the IL-1β/IL-10 ratio based ASD subgroups under cultures stimulated with food proteins (β-LG and gliadin) and candida proteins. Moreover, T cell cytokine production revealed close correlations with monocyte cytokine production, specific to each ASD subgroup. These findings support our hypothesis of sustained effects of maladapted TI on adaptive immunity in some ASD children.

The immune system operates with two components, innate and adaptive immunity. Innate immunity exerts rapid Ag-non-specific immune responses to contain infection until Ag-specific adaptive immunity sets in to clear hazards from the human body [26,27,28]. As opposed to adaptive immunity, innate immunity has been thought to lack lasting effects, since innate immune responses are Ag non-specific. However, for the past few years, it became clear that innate immunity can generate lasting effects or IIM by inducing metabolic and epigenetic changes in innate immune cells [29,30]. The key difference of IIM from adaptive immune memory is that the 2^nd^ stimuli to provoke IIM can be totally unrelated to the 1^st^ stimuli.

The presence of IIM was first convincingly shown as non-specific effects of BCG. Namely, BCG vaccination led to a greater reduction of infant mortality than predicted for decrease in mortality caused by tuberculosis in epidemiological and then randomized prospective studies [31,32]. These findings were further confirmed in rodent IIM models: BCG vaccination reduced from *Candida albicans* in the absence of lymphocyte that generates adaptive immune responses [33]. TI is the term that was given to describe these non-specific effects of IMM. Further studies revealed that TI can be generated in various innate immune cells including monocyte-macrophage lineage cells, natural killer (NK) cells [34], and bone marrow progenitor cells of innate immunity [35,36].

Metabolic and epigenetic changes have been extensively studied in ‘in vitro’ models of TI with the use of β-glucan derived from *Candida albicans*. Others have shown that β-glucan stimulates inflammasome through dectin-1, resulting in activation of signaling pathways involving Akt/PTEN/mTOR/HIF-1α [37]. This will shift oxidative phosphorylation (ATP synthesis) to glycolysis, resulting in decreased basal cellular respiration, and increased consumption of glucose, thereby leading to increase in lactate production [37]. These metabolic changes also affect synthesis of cholesterol and phospholipids [13,38], leading the replenishment of the Krebs cycle through formation of glutamate and α-keto-glutamate, and finally accumulation of fumarate [13,38]. A high concentration of fumarate hinders enzymatic actions of H3K4 demethylase, eventually resulting in epigenetic reprogramming [38,39,40].

Pre-administration of proinflammatory innate cytokines (IL-1, TNF-α, and IL-6) provided protection against a variety of microbes [41]. Among the cytokines administered, IL-1 showed superior effects over TNF-α or IL-6 [41]. In humans, TI associated protection has been mainly implicated with IL-1β and other IL-1 families [22]. Given the role of IL-1β in TI, excessive, dysregulated production of IL-1β is likely to cause maladapted TI, and resultant pathogenic consequences. Patients with gene mutations that lead to the over-production of IL-1β are known to reveal inflammatory symptoms involving multiple organs including the brain [22,42]. Moreover, chronic inflammatory conditions, including neuropsychiatric conditions have been implicated with maladapted TI [29,43].

As opposed to TI, tolerance is another form of IIM, causing lasting hypo-responsiveness against stimuli of innate immunity which is bested studied in endotoxin tolerance [44]. Upon induction of endotoxin tolerance, innate immune cells were reported to produce less inflammatory cytokines (TNF-α, IL-12, IL-6), but generate more counter-regulatory cytokines (IL-10 and TGF-β) with the subsequent endotoxin challenge [45,46]. LPS, a representative endotoxin, exerts its actions through TLR4, that results in activation of downstream signaling pathways the myeloid differentiation factor 88 (MyD88) and the TIR-domain-containing adaptor-inducing interferon-β (TRIF) [17]. Under the state of endotoxin tolerance, suppression of activation of these pathways are shown to be accomplished through multiple stages; down-regulation of TLR4, suppressed recruitment of MyD88 and TRIF to TLR4, reduced activation of IL-1 receptor-associated kinase (IRAK)1 and IRAK4, diminished activity of nuclear factor κ chain of B cell (NFκB), and up-regulated expression of counter-regulatory molecules such as SHIP1 (SH2 domain-containing inositol phosphatase 1) [47]. Induction of innate immune tolerance appears crucial in maintaining brain homeostasis; impaired innate immune tolerance has been implicated with pathogenesis of chronic neurodegenerative disorders including Alzheimer’s disease [29].

We have been analyzing monocyte functions in ASD subjects for the past 4–5 years, and we have reported variable differences in monocyte cytokine profiles in ASD subjects, which have been best reflected in IL-1β/IL-10 ratios [20]. This finding led us to subgroup ASD PBMo samples on the basis of IL-1β/IL-10 ratios generated by stimulating PBMo with a panel of TLR agonists [20]. The IL-1β/IL-10 ratio based subgrouping revealed notable changes in miRNA expression and mitochondrial respiration across the IL-1β/IL-10 based ASD subgroups [20]. In addition, we also found the IL-1β/IL-10 ratio based ASD subgroup specific associations between metabolic changes (mitochondrial respiration) and monocyte cytokine profiles [21]. This was also true when we examined correlations between serum miRNA levels and monocyte cytokine profiles in the IL-1β/IL-10 ratio based ASD subgroups (manuscript submitted for publication). In addition, it is our observation that the ASD subjects whose PBMo revealed low or high IL-1β/IL-10 ratios tend to reveal fluctuating behavioral symptoms following immune insults.

The above described findings indicate that the innate immune responses in the high and low IL-1β/IL-10 ratio ASD subgroups may be altered as a result of maladapted TI, leading to clinical pictures of chronic inflammation affecting multi-organs. To address this possibility, we assessed whether responses to β-glucan also differed in the IL-1β/IL-10 ratio based ASD subgroups. Our results revealed significant differences of IL-1β/IL-10 ratios and monocyte cytokine profiles across the IL-1β/IL-10 based ASD subgroups under β-glucan stimulated cultures (Table 5). If the previously observed changes of monocyte cytokine profile were only limited to TLR pathways, such changes would be more likely to be associated with defects specific to TLR mediated signaling pathways. β-glucan stimulates a connonical inflammasome pathway and amplifies signaling through the TLR pathway [25]. Thus, the fact that we observed similar changes under β-glucan stimulated cultures supports the role of maladapted TI in ASD subjects; TI has been reported in multiple signaling pathways associated with innate immunity [30].

This study also addressed how T cell cytokine profiles in response to recall Ags are associated with changes in innate immune responses in the IL-1β/IL-10 based ASD subgroups. TI induced excessive innate immune responses are likely to induce more exuberant adaptive responses, thereby increasing the risk of unwanted immune reactivity to benign environmental antigens such as food antigens. However, if chronic activation of adaptive immunity through maladapted TI is sustained, this may eventually lead to suppression of adaptive immunity, regaining immune homeostasis. Such secondary suppression of adaptive immunity could occur by clonal deletion or through actions exerted by regulatory T cells [48,49]. However, how dysregulated innate immune abnormalities (presumably maladapted TI) affect adaptive immune responses in ASD children is poorly understood. Since ASD subjects have a high frequency of GI symptoms and delayed type reactivity to common food proteins and candida proteins [50], we opted to check T cell cytokine profiles against representative food proteins and candida antigens, when assessing associations between T cell cytokine profiles and monocyte cytokine profiles.

With this analysis, we observed significant differences in the production of TNF-α and IL-10 in response to β-LG and gliadin between the IL-1β/IL-10 based ASD subgroups. Namely, the high ratio groups tended to reveal higher levels of these two cytokines than the other subgroups (Table 5). On the other hand, the normal and low ratio ASD subgroups revealed higher IFN-γ production in response to candida protein (Table 5). As for TNF-α in response to candida Ag, both the high and low IL-1β/IL-10 ratio subgroups revealed a higher TNF-α production (Table 5). In contrast, IL-10 production against candida Ag was lower in the ASD subgroups, with the normal ratio group being the lowest. These results indicate complex associations between adaptive immune responses and altered innate immune responses in the ASD subjects.

When correlations between T cell cytokine production against β-LG and gliadin and monocyte cytokine profiles were assessed, positive correlations between IFN-γ production and production of monocyte cytokines (IL-1β, IL-10, and IL-6) were predominantly observed in the normal ratio ASD subgroup (Table 6). In contrast, correlations between production of TNF-α, IL-10, and IL-12 by PBMCs in responses to these food proteins and monocyte cytokines were predominantly found in either the high ratio ASD subgroup (for TNF-α and IL-10), or the low ratio ASD subgroup (for IL-12) (Table 7, Table 8 and Table 9). Non-ASD controls revealed little correlations except for TNF-α production by PBMCs and TNF-α production by PBMo (Table 7). These results indicate that responses to benign environmental antigens, such as food proteins, appear to be differently affected by innate immune responses in the IL-1β/IL-10 based ASD subgroups, perhaps reflecting maladapted TI.

When correlations between T cell cytokine production against candida proteins and monocyte cytokine profiles were assessed, we observed similar results; predominant correlations with monocyte cytokines in the normal ratio group for IFN-γ production, with more predominant correlations in the high and/or low ratio ASD subgroups for TNF-α, IL-10, and IL-12 production by PBMCs (Table 6, Table 7, Table 8 and Table 9). Non-ASD controls only revealed positive correlations between IL-10 production by PBMCs and monocyte cytokine productions (Table 8). Since we were testing cellular immune reactivity to benign environmental Ags (food proteins and candida which is a normal component of the gut microbiome), a lack of close associations of inflammatory T cell cytokine profiles against these benign Ags in the non-ASD controls, and monocyte cytokine profiles may indicate the establishment of immune tolerance in adaptive immunity. Negative correlations between IL-10 production against candida protein and monocyte cytokine production may also indicate the establishment of immune homeostasis in non-ASD controls (Table 8). While the presence of close associations between monocyte cytokine profiles and T cell cytokine profiles (especially inflammatory cytokines—IFN-γ, TNF-α, and IL-12) in the ASD subgroups may indicate on-going effects of innate immunity in the absence of oral tolerance; this may also reflect maladapted TI.

When we assessed the clinical features in the IL-1β/IL-10 based ASD subgroups, we observed a high frequency of GI symptoms and a history of NFA with favorable responses to dietary interventions (usually gluten-free, dairy-free diet), consistent with our previous reports [20,21]. The IL-1β/IL-10 low ratio subgroup tended to have a higher frequency of history of NFA. β-GL is thought to be a major milk protein component associated with delayed type cellular immune reactivity in predisposed individuals. Interestingly, in the low IL-1β/IL-10 ratio ASD subgroup, IL-12 production by PBMCs in response to β-GL, a component was negatively correlated with counter-regulatory monocyte cytokines (IL-10, sTNFRII, and TGF-β) (Table 9). In contrast, IL-10 production in response to the β-GL was mostly positively correlated with monocyte cytokines production in the low ratio ASD subgroup (Table 9). This may indicate that altered innate immune responses may be regulating T cell cytokine production in the direction of suppressing excessive responses to recall antigens. On the other hand, in the high ratio group, there are positive correlations between IFN-γ, TNF-α, and IL-10 production in response to β-GL and monocyte cytokines (both inflammatory and counter-regulatory cytokines). In the high ratio group, such negative regulations may not be properly in place. Alternatively, with immune stimuli, such negative regulations may be easily lost secondary to maladapted TI.

## 4. Materials and Methods

### 4.1. Study Subjects

This study protocol was approved as protocols 15:45 (approval on 28 April 2016) and 17:53 (approval on 26 April 2018) by the Institutional Review Board, Saint Peter’s University Hospital, New Brunswick, NJ, USA. This study included both ASD and typically developing (TD), non-ASD control subjects. We obtained the signed consent forms prior to sample obtainment from parents or guardians when the study subjects were minor (<18 years of age) or were judged unable to give consent by him/herself due to intellectual disability. History of comorbidities including food allergy (FA), asthma, allergic rhinitis (AR), specific antibody deficiency (SAD), sleep disorder, and seizure disorders were assessed in ASD children by history taking and medical chart review. This study excluded subjects with chromosomal abnormalities, well defined gene mutations, or chronic diseases involving major organs, but did not exclude subjects with minor medical conditions highly prevalent in general population, such as seasonal allergy. In this study, both ASD and non-ASD, TD subjects were enrolled, and the signed consent forms were obtained prior to entering the study. Consent was obtained from parents/guardians if participant was a minor (<18 years old) or parents/guardians had custody. For ASD children, we also assessed whether they had a history of FA, asthma, AR, SAD, or seizure disorders. Subjects diagnosed with chromosomal abnormalities, other genetic diseases, or well characterized chronic medical conditions involving a major organ, were excluded from the study. Subjects with common minor medical conditions such as AR, mild to moderate asthma, eczema were not excluded from the study.

ASD subjects: ASD subjects (*n* = 152) were recruited from the Pediatric Allergy/Immunology Clinic. Diagnosis of ASD was made at various autism diagnostic centers, including ours. The ASD diagnosis was based on the Autism Diagnostic Observation Scale (ADOS) and/or Autism Diagnostic Interview-Revisited (ADI-R), and other standard measures. ASD subjects were also evaluated for their behavioral symptoms and sleep habits with the Aberrant Behavior Checklist (ABC) [51] and the Children’s Sleep Habits Questionnaires (CSHQ) [52], respectively. Information regarding cognitive ability and adaptive skills were obtained from previous school evaluation records performed within 1 year of enrollment in the study; these results were based on standard measures such as the Woodcock-Johnson III test (for cognitive ability), and Vineland Adaptive Behavior Scale (VABS) (for adaptive skills) [53].

Non-ASD controls: A total of 41 non-ASD subjects served as controls. These subjects were recruited in the Pediatrics Subspecialty and General Pediatrics Clinics at our institution. These subjects were typically growing and satisfied our exclusion criteria.

Table 1 reveals demographics of study subjects. Gender difference did not reveal significant changes regarding monocyte cytokine profiles and IL-1β/IL-10 ratios in both ASD and non-ASD groups, as reported before [21].

Diagnosis of FA: IgE mediated FA was diagnosed with reactions to offending food, by affecting the skin, GI, and/or respiratory tract immediately (within 2 h) after intake with positive prick skin testing (PST) reactivity, and/or presence of food allergen-specific serum IgE. Non IgE mediated FA (NFA) was diagnosed if GI symptoms resolved, following implementation of a restricted diet (i.e., avoidance of offending food), and symptoms recurred with re-exposure to offending food [54]. NFA was also defined as being non-reactive to PFT and negative for serum IgE specific for food allergens [54].

Diagnosis of asthma and AR: AR and allergic conjunctivitis (AC) were diagnosed when subjects had corresponding clinical features along with positive PST reactivity and/or positive serum IgE specific to [55,56]. Asthma was diagnosed following the asthma guidelines from the Expert Panel Report 3 [57].

Antibody deficiency syndrome: When the subject revealed protective levels of antibodies in less than 11 of 14 serotypes of *Streptococcus pneumonia* after the booster dose of Pneumovax^®^ or PCV13^®^, he/she was diagnosed with SAD [58]. Antibody (Ab) levels greater than 1.3 µg/mL were considered protective [58].

### 4.2. Sample Collection

Venous blood samples were obtained by physician in this study. We obtained one sample from each non-ASD control. As for ASD subjects, we obtained multiple blood samples from select ASD subjects (*n* = 22), in order to assess variability of T cell cytokine profiles. If parents or study subjects preferred, we applied a topical lidocaine/prilocaine cream (Emla cream^®^,IGI Laboratories, Buena, NJ, USA) to the site of venipuncture prior to blood sampling.

### 4.3. Cell Cultures

Ficoll–Hypaque density gradient centrifugation was used for separating PBMCs. From PBMCs, PBMo were further purified using magnetic beads labeled with anti-CD3, CD7, CD16, CD19, CD56, CD123, and glycophorin A (monocyte separation kit II—human, MILTENYI BIOTEC, Cambridge, MA, USA). Namely, this column depletes T, B, natural killer, and dendritic cells from PBMCs with combination of these antibodies.

Cytokine production by purified PBMo was induced by incubating cells overnight (2.5 × 10^5^ cells/mL) with a panel of agonist of toll like receptors (TLRs). This assay system was designed to reflect the effects of microbial byproducts commonly encountered in real life. Lipopolysaccharide (LPS), a TLR4 agonist, represents a signaling pathway activated in response to a Gram negative (G (–)) bacteria. Zymosan, a TLR2/6 agonist, mimics an innate activation signal in response to G (+) bacteria and fungi. CL097, a TLR7/8 agonist, activates innate signaling pathways in response to ssRNA viruses that cause common respiratory infection. We adapted these stimuli in our assay system, since these innate immune stimuli have been widely used by others. In addition, candida heat extract as a source of β-glucan, dectin-1 agonist, was used as well as a representative C-lectin receptor agonist. PBMos were incubated overnight with LPS (0.1 µg/mL, GIBCO-BRL, Gaithersburg, MD, USA), zymosan (50 µg/mL, Sigma-Aldrich, St. Luis, Mo, USA), C097 (water-soluble derivative of imidazoquinoline, 20 µM, InvivoGen, San Diego, CA, USA), and candida heat extract (HCKA, heat killed candida albicans (10^7^ cells/mL, InVivogen, San Diego, CA) as a source of β-glucan, a dectin 1 agonist, in RPMI 1640 with additives as previously described [59]. Overnight incubation (16–20 h) was adequate to induce the optimal responses in this setting. The culture supernatant was used for cytokine assays.

Production of T cell cytokines (IFN-γ, IL-5, TNF-α, IL-10, IL-12p40, IL-17, and TGF-β) was assessed by incubating PBMCs (10^6^ cells/mL) with representative recall antigens including β-lactoglobulin (β-LG, 10 µg/mL, Sigma Aldrich), soy protein (5 µg/mL, Ross, Nutley, NJ, USA), gliadin (10 µg/mL, Sigma-Aldrich), milk protein (100 µg/mL, Ross), and candida protein (5 µg/mL, Greer, Lenoir, NC, USA) for 4 days in RPMI1540 with additives as reported previously [60]. As noted in our previous study, a 4 day incubation period resulted in the optimal production of these cytokines in this culture setting.

Levels of C-C chemokine ligand 2 (CCL2), CCL7, interferon-γ (IFN-γ), IL-1β, IL-5, IL-6, IL-10, IL-12p40, IL-17, transforming growth factor-β (TGF-β), tumor necrosis factor-α (TNF-α), and soluble TNF receptor II (sTNFRII) cytokines were measured by enzyme-linked immuno-sorbent assay (ELISA); 10–100 µL/well supernatants were used for ELISA. The OptEIA™ Reagent Sets (BD Biosciences, San Jose, CA, USA) were used for ELISA of IFN-γ, IL-1β, IL-6, IL-10, IL-12p40, and TNF-α. For CCL2, CCL7, IL-17 (IL-17A), sTNFRII, and TGF-β ELISA, reagents were obtained from BD Biosciences and R & D (Minneapolis, MN, USA). IL-23 ELISA kit was purchased from eBiosciences, San Diego, CA. Intra- and inter-variations of cytokine levels were less than 5%.

### 4.4. Categorizing ASD Samples on the Basis of IL-1β/IL-10 ratios

Previously, we reported that changes in the IL-1β/IL-10 ratios best reflect altered cytokine profiles and miRNA expression by PBMo [20]. We divided ASD samples into subgroups based on the IL-1β/IL-10 ratios produced by ASD PBMo, following the criteria used in our previous study [20], as outlined below. In this study, we used IL-1β/IL-10 ratios generated under cultures with medium only, LPS, zymosan, CL097, and candida heat extract as a source of β-glucan.

High IL-1β/IL-10 ratio subgroup: In this subgroup, ASD PBMo showed 2 standard deviation (SD) higher IL-1β/IL-10 ratios under at least one culture condition and/or 1 SD greater IL-1β/IL-10 ratios under more than two culture conditions, as compared to control PBMo.

Normal IL-1β/IL-10 ratio subgroup: ASD PBMo revealed IL-1β/IL-10 ratios less than +1 SD, and greater than –1 SD under all the culture conditions, or IL-1β/IL-10 ratios greater than 1 SD but less than 2 SD under only one culture condition, as compared to control PBMo.

Low IL-1β/IL-10 ratio subgroup: ASD PBMo revealed IL-1β/IL-10 ratios less than –1 SD under at least one culture condition, as compared to control PBMo.

Among 25 ASD subjects in whom we obtained blood samples at 2–4 time points, most subjects revealed that blood samples from these subjects were categorized in the same group except for six ASD subjects. PBMo from these six ASD subjects showed high ratios at 1–2 time points and normal/low ratios at one time point. Eleven ASD subjects revealed low ratios at 1–2 time points and normal ratios at one time point. When comparing their clinical characteristics, they were categorized as high and low IL-1β/IL-10 ratio groups, respectively.

### 4.5. Statistical Analysis

We used a two tailed Mann–Whitney test for comparison of two sets of numerical data. Comparison of multiple data sets was assessed by one-way ANOVA and/or Kruskal—Wallis test. Normality of the numerical data were assessed by skewness and kurtosis (Omnibus) with α = 0.2. When assessing differences in frequency between two groups, we used the Fisher exact test. For assessing differences in frequency among multiple groups, we used the Chi-Square test and the Likelihood ratio. A linear association between two data sets was determined by Spearman test. A *p* value of less than 0.05 was considered nominally significant. Co-variance analysis was done with the use of general linear model for a fixed factor or for a variable factor. For assessing longitudinal changes, repeated measures of analysis of variance were used. NCSS19 (NCSS, LLC. Kaysville, UT, USA) was used for statistical analysis.

### 4.6. Availability of Data and Material

Clinical features of the ASD are available through NDAR data base (https://ndar.nih.gov/). The additional datasets used and/or analyzed during the current study are available from the corresponding author on reasonable request.

## 5. Conclusions

This study revealed similar changes in IL-1β/IL-10 ratios and monocyte cytokine profiles under β-glucan stimulated cultures as observed in the cultures stimulated with TLR agonist, indicating that changes in innate immune responses are not limited to TLR pathways. Correlations between the changes in monocyte cytokine profiles and T cell cytokine profiles in response to benign environmental Ags differed across the IL-1β/IL-10 based ASD subgroups. Such correlations were much less evident in non-ASD controls, supporting possible effects of maladapted TI in some ASD children. Further analysis of the status of innate immunity in association with TI or tolerance will be helpful in understanding the inflammatory subtype of ASD.

## Figures and Tables

**Table 1 ijms-20-04731-t001:** Demographics and clinical features in the IL-1β/IL-10 based autism spectrum disorder (ASD) subgroups.

	**IL-1β/IL-10 Ratio Based ASD Subgroups**		
	**High (*n* = 65)**	Normal (*n* = 51)	Low (*n* = 36)
Age (year) median (range)	9.9 (2.5–26.3)	8.5 (2.2–24.5)	11.4 (2.6–22.4)
Age (year) Mean ± SD ^1^	11.3 ± 5.8	10.6 ± 6.1	12.1 ± 5.6
Gender (M:F)	52:13 (80.0%:20.0%)	46:5 (80.4%:19.6%)	32:4 (88.9%:11.1%)
Ethnicity	AA 5. Asian 17, W 40, Mixed 5	AA 2, Asian 12, W 35, mixed 2	AA 4, Asian 5, W 27,
**Clinical Characteristics ^2^**			
ASD severity			
Severe	39/65 (60.0%)	31/51 (60.8%)	20/36 (55.6%)
Moderate	13/65 (20.0%)	9/51(17.6%)	9/36 (25.0%)
Mild	13/65 (20.0%)	11/51(21.6%)	7/36 (19.4%)
Cognitive development (<1st %)	39/65 (60.0%)	38/51 (74.5%)	25/36 (69.4%)
Disturbed sleep	21/65 (32.3%)	16/51 (33.3%)	12/36 (33.3%)
GI symptoms	46/65 (70.8%)	36/51 (70.6%)	29/36 (80.6%)
History of NFA	30/65 (46.2%)	22/51 (43.1%)	22/36 (61.1%)
Seizure disorders	7/65 (10.8%)	8/51 (8.3%)	3/36 (8.3%)
Specific antibody deficiency	11/65 (16.9%)	7/51 (13.75)	8/36 (22.2%)
Allergic rhinitis	15/65 (23.1%)	10/51 (19.6%)	5/36 (13.9%)
Asthma	8/65 (12.3%)	4/51 (7.8%)	4/36 (11.1%)

^1^ Abbreviations used: AA, African Americans; ADHD, attention deficit hyperactivity disorder; AEDs, anti-epileptic drugs; F, female; GI, gastrointestinal; M, male; NFA, non-IgE mediated food allergy; SD, standard deviations; SSRIs, selective serotonin receptor inhibitors; W, Caucasians. ^2^ No significant differences in frequency of clinical characteristics in the IL-1β/IL-10 ratio based ASD subgroup by Chi-Square test (*p* > 0.05).

**Table 2 ijms-20-04731-t002:** Frequency of medication use in the IL-1β/IL-10 based ASD subgroups.

	IL-1β/IL-10 Ratio Based ASD Subgroups (Sample Numbers)		
	High (*n* = 70) ^1^	Normal (*n* = 74)	Low (*n* = 43)
**Medication Use ^2^**			
Neuroleptics	6/70 (8.6%)	12/74 (15.7%)	6/43 (14.6 %)
ADHD medications	11/70 (15.7%)	6/74 (8.1%)	8/43 (18.6%)
AEDs	12/70 (17.1%)	13/74 (17.6%)	6/43 (14.6 %)
SSRIs	13/70 (18.5%)	11/74 (14.9%)	8/43 (18.6%)

^1^ Frequency of medication use at the time of sample obtainment. It should be noted that samples were obtained at 2–4 time points in a total of 25 ASD subjects. Frequency of medication use did not significantly differ in the IL-1β/IL-10 based ASD subgroups by Chi-Square test (*p* > 0.05). ^2^ Medications on which study subjects were on when blood samples were obtained.

**Table 3 ijms-20-04731-t003:** Changes in IL-1β/IL-10 ratios and monocyte production in the IL-1β/IL-10 based ASD subgroups under β-glucan stimulated culture conditions.

Cytokines Produced	IL-1β/IL-10 Ratio Based ASD Subgroups			Non-ASD	*p*-Value ^4^
	High (*n* = 70) ^1^	Normal (*n* = 74)	Low (*n* = 43)	Controls (*n* = 41)	
**Without Stimuli (Medium Only) ^6^**					
IL-1β/IL-10	1.44 ± 1.9 ^2,6^	0.71 ± 0.48	0.42 ± 0.29	1.02 ± 2.06	0.00000
IL-1β	208.8 ± 233.1 ^3^	201.2 ± 236.9	101.6 ± 93.9 ^6^	260.8 ± 317.5 ^5^	0.00612
IL-10	212.3 ± 170.9	248.6 ± 211.7	299.2 ± 271.7	436.4 ± 837.0 ^5^	0.09448
TNF-α	76.2 ± 136.0 ^6^	42.9 ± 116.8	15.7 ± 26.0	81.9 ± 121.1	0.00006
sTNFRII	301.1 ± 167.9	370.5 ± 187.2	340.8 ± 169.8	331.3 ± 200.8	0.19151
TGFβ	577.4 ± 321.0 ^6^	370.5 ± 187.2	691.3 ± 328.8	650.6 ± 328.9	0.00703
CCL2	18,423 ± 7617	18,519 ± 6259	17,637 ± 8364	18,554 ± 7775	0.90492
**With β-Glucan**					
IL-1β/IL-10	15.4 ± 20.6 ^6^	4.1 ± 2.1	2.7 ± 1.7	4.5 ± 3.1	0.00000
IL-1β	2355.1 ± 1077.5 ^6^	2266.6 ± 1083.6	1541.8 ± 776.9 ^6^	1812.3 ± 1028.3	0.00008
IL-10	266.4 ± 205.7 ^6^	670.7 ± 394.8	806.5 ± 469.4	517.7 ± 395.8	0.00000
TNF-α	1456.6 ± 1125.3 ^6^	1096.7 ± 967.0	740.1 ± 484.4 ^6^	1138.6 ± 876.1	0.00761
sTNFRII	426.7 ± 227.1 ^6^	722.2 ± 464.5	825.1 ± 518.9	595.8 ± 567.9	0.00000
TGFβ	439.6 ± 298.0 ^6^	588.4 ± 298.3	598.3 ± 259.3	560.8 ± 298.5	0.00312
CCL2	4640 ± 3741 ^6^	7308 ± 5596	10001 ± 7502 ^6^	6807 ± 6568	0.00000

^1^ Total sample numbers in the group; ^2^ The results are expressed as mean values ± SD. ^3^ Cytokine concentrations are shown as pg/mL; ^4^
*p* values by one-way ANOVA with α = 0.2 (Terry–Hoeffding test). Comparisons between groups by ANOVA with α = 0.05. ^5^ Values from non-ASD controls differed from ASD subgroups. ^6^ Differed from other ASD subgroups by ANOVA (α = 0.05).

**Table 4 ijms-20-04731-t004:** T cell cytokine production by peripheral blood mononuclear cells (PBMCs) in response to β-LG, gliadin, or candida Ag in the IL-1β/IL-10 based ASD subgroups.

Cytokines	IL-1β/IL-10 Ratio Based ASD Subgroups			Non-ASD	*p*-Value ^3^
	High (*n* = 70) ^1^	Normal (*n* = 74)	Low (*n* = 43)	Controls (*n* = 41)	
IFN-γ					
medium only	54.5 ± 155.5 ^2^	65.1 ± 183.9	71.1 ± 134.6	28.7 ± 39.5	0.3431
β-lactoglobulin	152.4 ± 229.9	112.7 ± 181.6	178.6 ± 291.7	77.8 ± 122.9	0.3486
gliadin	107.4 ± 179.3	94.2 ± 192.6	114.3 ± 212.7	43.5 ± 56.1	0.2129
Candida	189.4 ± 290.9	278.9 ± 442.7	265.0 ± 351.3	172.3 ± 342.1 ^4^	0.0119
TNF-α					
medium only	31.2 ± 136.9	25.2 ± 130.1	61.7 ± 177.5	0.9 ± 2.4	0.0309
β-lactoglobulin	230.7 ± 219.2 ^5^	137.3 ± 201.0	108.8 ± 191.2	154.0 ± 196.8	0.00043
gliadin	220.9 ± 219.4 ^5^	125.1 ± 193.0	90.5 ± 173.4	112.9 ± 159.0	0.00002
Candida	64.6 ± 219.2	38.3 ± 139.2	58.8 ± 183.3	28.7 ± 85.5	0.01695
IL-10					
medium only	60.4 ± 161.3	38.5 ± 98.3	80.6 ± 164.1	30.5 ± 63.3	0.107
β-lactoglobulin	1358.5 ± 523.7 ^5^	1146.4 ± 495.1	1044.3 ± 526.6	1119.2 ± 423.1	0.0073
gliadin	808.3 ± 371.0 ^5^	760.4 ± 446.8	572.1 ± 387.7 ^5^	742.5 ± 374.7	0.0063
Candida	66.6 ± 91.0	41.5 ± 60.0	60.4 ± 90.1	93.6 ± 152.0 ^4^	0.9926

^1^ Total sample numbers in each study group. ^2^ The results are expressed as mean values ± SD. Cytokine concentrations are shown as pg/mL; ^3^
*p* values by one-way ANOVA with α = 0.2 (Terry–Hoeffding test). Comparisons between groups by ANOVA with α = 0.05. ^4^ Values from non-ASD controls differed from ASD subgroups. ^5^ Differed from other ASD subgroups by ANOVA (α = 0.05).

**Table 5 ijms-20-04731-t005:** Correlations between IFN-γ production by PBMCs and monocyte cytokine production.

Correlation Coefficient	IL-1β/IL-10 Ratio Based ASD Subgroups			Non-ASD
	High (*n* = 70) ^1^	Normal (*n* = 74)	Low (*n* = 43)	Controls (*n* = 41)
**In Response to β-Lactoglobulin**				
IL-1β/IL-10 ratio				
No stimulant ^3^	--	--	0.3236 (*p* < 0.05)	--
LPS	--	0.3084 (*p* < 0.02) ^2^	--	--
CL097	--	−0.2662 (*p* < 0.05)	--	--
IL-1β				
No stimulant	0.2668 (*p* < 0.05)	---	0.3941 (*p* < 0.02)	--
LPS	0.3141 (*p* < 0.02)	0.3663 (*p* < 0.005)	--	--
CL097	---	−0.2714 (*p* < 0.05)	0.3912 (*p* < 0.02)	--
IL-10				
No stimulant	0.332 (*p* < 0.01)	--	0.3258 (*p* < 0.05)	--
Zymosan	--	0.2439 (*p* < 0.05)	--	--
CL097	--	--	--	−0.3586 (*p* < 0.05)
IL-6				
No stimulant	0.2917 (*p* < 0.05)	0.2744 (*p* < 0.05)	--	0.3351 (*p* < 0.05)
LPS	--	0.3216 (*p* < 0.01)	--	--
sTNFRII				
No stimulant	−0.2524 (*p* < 0.05)	--	--	--
β-glucan	−0.3615 (*p* < 0.005)	--	--	--
CCL2				
No stimulant	--	0.2928 (*p* < 0.02)	--	--
β-glucan	--	0.2511 (*p* < 0.05)	--	--
**In Response to Gliadin**				
IL-1β/IL-10 ratio				
No stimulant	--	--	0.3772 (*p* < 0.02)	--
LPS	--	0.3843 (*p* < 0.005)	--	--
IL-1β				
No stimulant	0.2713 (*p* < 0.05)	--	0.4462 (*p* < 0.005)	--
LPS	--	0.481 (*p* < 0.0001)	--	--
IL-10				
No stimulant	0.2748 (*p* < 0.05)	--	--	--
LPS	--	0.3413 (*p* < 0.005)	--	--
TNF-α				
No stimulant	0.3398 (*p* < 0.01)	--	--	--
Zymosan	--	--	--	−0.326 (*p* < 0.05)
IL-6				
No stimulant	0.2621 (*p* < 0.05)	0.2426 (*p* < 0.05)	--	--
LPS	--	0.396 (*p* < 0.001)	--	--
Zymosan	--	0.3036 (*p* < 0.02)	--	--
sTNFRII				
Zymosan	−0.3397 (*p* < 0.01)	--	--	--
β-glucan	−0.4695 (*p* < 0.0001)	--	--	--
**In Response to Candida Protein**				
IL-1β/IL-10 ratio				
No stimulant	--	−0.324 (*p* < 0.01)	--	--
IL-1β				
No stimulant	--	0.2607 (*p* < 0.05)	--	--
Zymosan	--	−0.2677 *p* < 0.05)	--	--
IL-10				
No stimulant	--	0.2567 (*p* < 0.05)	--	--
IL-6				
No stimulant	--	0.379 (*p* < 0.001)	--	--
Zymosan	--	0.2932 (*p* < 0.02)	--	--
β-glucan	--	0.2507 (*p* < 0.05)	--	--
sTNFRII				
No stimulant	--	--	--	−0.3515 (*p* < 0.05)
LPS	--	--	--	−0.346 (*p* < 0.05)
Zymosan	--	--	--	−0.3768 (*p* < 0.05)
CCL2				
No stimulant	--	0.3071 (*p* < 0.01)	--	--
Zymosan	--	0.2385 (*p* < 0.05)	--	--
CL097	--	0.3292 (*p* < 0.005)	--	--
β-glucan	--	0.277 (*p* < 0.05	--	−0.4391 (*p* < 0.01)

^1^ Total sample numbers in each study group; ^2^ Correlation coefficient by Spearman test with *p* value shown in ( ) when *p* value is at least *p* < 0.05. ^3^ Culture conditions monocyte cytokines produced under.

**Table 6 ijms-20-04731-t006:** Correlations between TNF-α production by PBMCs and monocyte cytokine production.

Correlation Coefficient	IL-1β/IL-10 Ratio Based ASD Subgroups			Non-ASD
	High (*n* = 70) ^1^	Normal (*n* = 74)	Low (*n* = 43)	Controls (*n* = 41)
**In Response to β-Lactoglobulin**				
IL-1β/IL-10 ratio				
LPS ^3^	--	--	--	0.3541 (*p* < 0.05)
CL097	--	−0.2744 (*p* < 0.05) ^2^	0.3916 (*p* < 0.02)	--
IL-1β				
LPS	0.2858 (*p* < 0.05)	--	--	0.3411 (*p* < 0.05)
CL097	0.4575 (*p* < 0.0005)	--	0.4069 (*p* < 0.02)	0.5101 (*p* < 0.005)
IL-10				
No stimulant	0.3339 (*p* < 0.01)	--	--	--
LPS	0.2858 (*p* < 0.05)	--	--	--
CL097	--	0.253 (*p* < 0.05)	--	--
TNF-α				
No stimulant	0.4883 (*p* < 0.0001)	--	--	0.569 (*p* < 0.0002)
LPS	0.6857 (*p* < 0.0001)	--	--	0.5103 (*p* < 0.002)
Zymosan	0.4558 (*p* < 0.0005)	--	--	0.3469 (*p* < 0.05)
CL097	0.5481 (*p* < 0.0001)	--	0.3364 (*p* < 0.05)	0.4643 (*p* < 0.005)
β-glucan	0.5469 (*p* < 0.0001)	--	--	--
IL-6				
No stimulant	0.3887 (*p* < 0.02)	--	--	--
LPS	0.2776 (*p* < 0.05)	--	0.3197 (*p* < 0.05)	0.4241 (*p* < 0.01)
sTNFRII				
LPS	--	--	--	−0.4695 (*p* < 0.005)
Zymosan	--	--	--	−0.3532 (*p* < 0.05)
CL097	--	0.2632 (*p* < 0.05)	−0.3166 (*p* < 0.05)	−0.4571 (*p* < 0.005)
β-glucan	--	--	--	--
CCL2				
β-glucan	--	--	--	−0.4411 (*p* < 0.01)
**In Response to Gliadin**				
IL-1β/IL-10 ratio				
LPS	--	0.2515 (*p* < 0.05)	--	--
CL097	--	−0.3013 (*p* < 0.02)	0.3561 (*p* < 0.05)	--
IL-1β				
LPS	0.3098 (*p* < 0.02)	0.2441 (*p* < 0.05)	--	--
CL097	0.2944 (*p* < 0.02)	--	0.4395 (*p* < 0.005)	--
β-glucan	--	0.2633 (*p* < 0.05)	--	--
IL-10				
No stimulant	0.2823 (*p* < 0.05)	--	--	--
LPS	0.3098 (*p* < 0.02)	--	--	--
CL097	02917 (*p* < 0.02)	--	0.3642 (*p* < 0.02)	--
TNF-α				
No stimulant	0.5168 (*p* < 0.0001)	--	0.5156 (*p* < 0.001)	0.4179 (*p* < 0.01)
LPS	0.6777 (*p* < 0.0001)	--	0.3462 (*p* < 0.02)	0.452 (*p* < 0.005)
Zymosan	0.4567 (*p* < 0.0001)	--	--	--
CL97	0.4268 (*p* < 0.0005)	--	0.4201 (*p* < 0.01)	0.5626 (*p* < 0.001)
β-glucan	0.4778 (*p* < 0.0001)	--	--	--
IL-6				
CL097	0.2612 (*p* < 0.05)	--	0.3937 (*p* < 0.02)	0.3695 (*p* < 0.05)
β-glucan	--	--	--	0.3939 (*p* < 0.02)
sTNFRII				
CL097	--	0.2481 (*p* < 0.05)	--	--
CCL2				
β-glucan	−0.2509 (*p* < 0.05)	--	--	−0.327 (*p* < 0.05)
**In Response to Candida Protein**				
IL-1β/IL-10 ratio				
No stimulant	−0.3303 (*p* < 0.01)	--	--	--
CL097	--	--	0.5351 (*p* < 0.005)	--
IL-1β				
Zymosan	--	--	−0.4518 (*p* < 0.005)	0.3507 (*p* < 0.05)
CL097	0.4228 (*p* < 0.0005)	--	0.5832 (*p* < 0.0001)	--
β-glucan	--	--	−0.3169 (*p* < 0.05)	--
IL-10				
CL097	0.2486 (*p* < 0.05)	--	0.4879 (*p* < 0.001)	--
sTNFRII				
CL097	0.4237 (*p* < 0.0005)	--	0.5626 (*p* < 0.0001)	--
β-glucan	0.2814 (*p* < 0.02)	--	--	--
IL-6				
CL097	0.3848 (*p* < 0.005)	--	0.5636 (*p* < 0.0001)	--
CCL2				
CL097	--	--	--	−0.3362 (*p* < 0.05)

^1^ Total sample numbers in each study group; ^2^ Correlation coefficient by Spearman test with *p* value shown in ( ) when *p* value is at least *p* < 0.05. ^3^ Culture conditions monocyte cytokines produced under.

**Table 7 ijms-20-04731-t007:** Correlation between IL-10 production by PBMCs and monocyte cytokine production.

Correlation Coefficient	IL-1β/IL-10 Ratio Based ASD Subgroups			Non-ASD
	High (*n* = 70) ^1^	Normal (*n* = 74)	Low (*n* = 43)	Controls (*n* = 41)
**In Response to β-Lactoglobulin**				
IL-1β/IL-10 ratio			−0.3223 (*p* < 0.05) 0.3407 (*p* < 0.05)	
Zymosan ^3^	−0.3281 (*p* < 0.01) ^2^	--		--
CL097	--	--		--
IL-1β				
LPS	0.4682 (*p* < 0.0001)	0.3497 (*p* < 0.005)	--	--
CL097	0.3247 (*p* < 0.02)	--	0.4745 (*p* < 0.005)	--
IL-10				
No stimulant	0.4606 (*p* < 0.0005)	--	--	--
LPS	0.3998 (*p* < 0.005)	0.5026 (*p* < 0.0001)	0.3545 (*p* < 0.05)	0.5166 (*p* < 0.005)
Zymosan	0.2924 (*p* < 0.05)	--	--	--
CL097	--	--	0.3848 (*p* < 0.02)	--
TNF-α				
No stimulant	0.3132 (*p* < 0.02)	--	--	--
LPS	0.5669 (*p* < 0.0001)	0.3322 (*p* < 0.01)	--	--
CL097	0.3086 (*p* < 0.02)	--	0.4028 (*p* < 0.02)	--
IL-6				
LPS	--	0.2552 (*p* < 0.05)	--	--
CL097	--	--	0.4028 (*p* < 0.02)	--
**In Response to Gliadin**				
IL-1β/IL-10 ratio				
LPS	--	0.2504 (*p* < 0.05)	--	--
Zymosan	−0.252 (*p* < 0.05)	−0.2656 (*p* < 0.05)	--	--
CL097	--	--	0.3138 (*p* < 0.05)	--
IL-1β				
LPS	0.3834 (*p* < 0.005)	0.3866 (*p* < 0.005)	--	--
CL097	--	--	0.4537 (*p* < 0.005)	--
β-glucan	--	0.2576 (*p* < 0.05)	--	--
IL-10				
No stimulant	0.3233 (*p* < 0.01)	0.2398 (*p* < 0.05)	--	--
LPS	--	0.4354 (*p* < 0.0005)	0.3524 (*p* < 0.05)	--
Zymosan	0.3459 (*p* < 0.005)	--	--	--
CL097	--	--	0.3962 (*p* < 0.02)	--
β-glucan	0.2745 (*p* < 0.05)	--	--	--
TNF-α				
No stimulant	--	--	0.3378 (*p* < 0.05)	--
LPS	0.3368 (*p* < 0.01)	--	--.	--
Zymosan	0.285 (*p* < 0.05)	--	--	--
CL97	--	--	0.3422 (*p* < 0.05)	--
IL-6				
CL097	--	--	0.3235 (*p* < 0.05)	--
β-glucan	--	--	--	0.3374 (*p* < 0.05)
sTNFRII				
β/glucan	--	--	−0.3227 (*p* < 0.05)	--
**In Response to Candida Protein**				
IL-1β/IL-10 ratio				
No stimulant	−0.2785 (*p* < 0.05)	--	--	--
zymosan	−0.3022 (*p* < 0.02)	--	--	--
IL-1β				
CL097	0.2382 (*p* < 0.05)	--	--	--
IL-10				
LPS	0.2709 (*p* < 0.05)	--	--	--
β-glucan	--	--	--	−0.323 (*p* < 0.05)
sTNFRII				
Zymosan	--	0.2433 (*p* < 0.05)	--	0.3507 (*p* < 0.05)
CL097	0.2969 (*p* < 0.02)	--	--	0.3165 (*p* < 0.05)
β-glucan	--	−0.2419 (*p* < 0.05)	--	0.3342 (*p* < 0.05)
IL-6				
CL097	0.3021 (*p* < 0.05)	--	--	--
β-glucan	0.261 (*p* < 0.05)	--	--	--
sTNFRII				
Zymosan	--	--	--	−0.3154 (*p* < 0.05)
CCL2				
No stimulant	--	--	0.3198 (*p* < 0.05)	--
Zymosan	--	--	--	−0.3658 (*p* < 0.05)
CL097	--	--	--	−0.3416 (*p* < 0.05)
β-glucan	--	--	--	−0.5169 (*p* < 0.001)

^1^ Total sample numbers in each study group; ^2^ Correlation coefficient by Spearman test with *p* value shown in ( ) when *p* value is at least *p* < 0.05. ^3^ Culture conditions monocyte cytokines produced under.

**Table 8 ijms-20-04731-t008:** Correlation between IL-12 production by PBMCs and monocyte cytokine production.

Correlation Coefficient	IL-1β/IL-10 Ratio Based ASD Subgroups			Non-ASD
	High (*n* = 70) ^1^	Normal (*n* = 74)	Low (*n* = 43)	Controls (*n* = 41)
**In Response to β-Lactoglobulin**				
IL-1β/IL-10 ratio				
No stimulant ^3^	--	--	0.3867 (*p* < 0.02) ^2^	--
IL-1β				
β-glucan	--	--	−0.3908 (*p* < 0.02)	--
IL-10				
Zymosan	−0.2832 (*p* < 0.02)	--	−0.4993 (*p* < 0.005)	0.338 (*p* < 0.05)
TNF-α				
β-glucan	--	--	−0.345 (*p* < 0.05)	--
IL-6				
LPS	0.2796 (*p* < 0.05)	--	--	--
Zymosan	--	--	−0.3165 (*p* < 0.05)	--
sTNFRII				
No stimulant	−0.3463 (*p* < 0.01)	--	--	--
LPS	--	--	−0.3762 (*p* < 0.02)	−0.342 (*p* < 0.05)
Zymosan	--	--	−0.4228 (*p* < 0.01)	−0.3346 (*p* < 0.05)
β-glucan	--	--	−0.4733 (*p* < 0.005)	--
TGF-β				
No stimulant	--	--	−0.4374 (*p* < 0.01)	--
LPS	--	--	−0.4476 (*p* < 0.005)	--
Zymosan	--	--	−0.4546 (*p* < 0.005)	--
CL097	--	--	−0.4602 (*p* < 0.005)	--
β-glucan	--	--	−0.4356 (*p* < 0.01)	--
**In Response to Gliadin**				
IL-1β/IL-10 ratio				
No stimulant	--	--	--	0.3964 (*p* < 0.02)
IL-1β			‘	
β-glucan	--	0.2835 (*p* < 0.02)	--	--
IL-10				
Zymosan	−0.2667 (*p* < 0.05)	--	−0.3707 (*p* < 0.02)	--
TNF-α				
No stimulant	--	--	0.468 (*p* < 0.005)	--
LPS	--	0.255 (*p* < 0.05)	0.3284 (*p* < 0.05)	--
IL-6				
CL097	--	--	0.3248 (*p* < 0.05)	--
β-glucan	--	−0.2472 (*p* < 0.05)	--	--
sTNFRII				
No stimulant	−0.2898 (*p* < 0.02)	--	--	--
zymosan	−0.2466 (*p* < 0.05)	--	--	--
β-glucan	--	--	−0.4348 (*p* < 0.005)	--
TGF-β				
Zymosan	--	--	−0.3163 (*p* < 0.05)	--
β-glucan	--	--	−0.3341 (*p* < 0.05)	--
CCL2				
No stimulant	--	--	0.3084 (*p* < 0.05)	--
**In Response to Candida Protein**				
IL-1β/IL-10 ratio				
Zymosan	−0.3644 (*p* < 0.005)	--	--	--
CL097	--	--	--	−0.3143 (*p* < 0.05)
IL-10				
CL097	--	--	0.3205 (*p* < 0.05)	--
TNF-α				
CL097	--	--	0.3319 (*p* < 0.05)	--
TGF-β				
Zymosan	--	--	−0.3077 (*p* < 0.05)	--
CCL2				
No stimulant	−0.2456 (*p* < 0.05)	--	0.506 (*p* < 0.0005)	--
Zymosan	--	--	0.3125 (*p* < 0.05)	--
CL097	--	--	0.3431 (*p* < 0.05)	--

^1^ Total sample numbers in each study group; ^2^ Correlation coefficient by Spearman test with *p* value shown in ( ) when *p* value is at least *p* < 0.05. ^3^ Culture conditions monocyte cytokines produced under.

**Table 9 ijms-20-04731-t009:** Demographics of ASD subjects and non-ASD controls.

	ASD ^1^ Subjects (*n* = 152)	Non-ASD Controls (*n* = 41)
Age (years) ^2^		
median, range	10.0 (2.2–26.3)	11.2 (1.9–29.6)
average ± SD	11.2 ± 5.8	13.3 ± 8.0
Gender (M:F)	130:22 (85.5 %:14.5%)	24:17 (58.5%:41.5%)
Ethnicity	AA 11, Asian 34, Mixed 5, W 102	Asian 4, Mixed 6, W 31

^1^ Abbreviations used; AA; African American, ASD; autism spectrum disorder, F; female, GI; gastrointestinal, M; male, NFA: non-IgE mediated food allergy, SD; standard deviation, W; Caucasian. ^2^ Age at the time of study enrollment is shown.

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
