# Peer review of "Associations between Monocyte and T Cell Cytokine Profiles in Autism Spectrum Disorders: Effects of Dysregulated Innate Immune Responses on Adaptive Responses to Recall Antigens in a Subset of ASD Children"

_ijms, 2019, doi:10.3390/ijms20194731_

Round 1
Reviewer 1 Report
This is an interesting study addressing selectively T and monocyte immune responses of patients with ASD. Three groups of ASD were selected on the basis that patients were presenting a different initial inflammatory status as judged by the level of a canonical proinflammatory cytokine (IL1 beta) and the anti-inflammatory cytokine (IL10). PBMCs were challenged with different agonists of pattern recognition receptors to address specifically the mechanisms of the trained immunity. Several criticisms need addressing.
Table 3: It is important that the authors a.indicate the levels of the different cytokines when the cells are left in control conditions (in absence of stimulation) and b.compare these values between the different subgroups. It is indeed important to address whether activation status of the PBMCs is already different between the different subgroups. The authors in the table’s legend are alluding to ‘changes ‘ but we need to have an indication of the baseline levels to analyze indeed these changes caused by beta glucan stimulation Table 3: It is not clear what the P value is referring to and this need to be clarified while comparing the different subgroups versus NON ASD controls Point 1 also applies for the results presented in Table 4 while addressing the T cell cytokine response The expression levels of IL10 are very different while comparing the data from Table 3 and Table 4 and yet in response to the same treatment by candida albicans. How can this be explained? Please clarify.Author Response
Table 3: the results of monocyte cytokine production in the absence of stimuli were added to the revised Table 3.
Table 4: the results of T cell cytokine production in the absence of stimuli were added to the revised Table 4.
Differences in IL-10 production in Tables 3 and 4: Differences are expected to occur, since Table 3 presents the data of monocyte cytokine production by purified peripheral blood monocytes cultured overnight and Table 4 presents the data of T cell cytokine production by peripheral blood mononculear cells cultured for 4 days with recall antigens.
Reviewer 2 Report
While the authors report some interesting findings, there are several major limitations to the statistical analyses performed that limit clear conclusions at present. Notably, for Tables 3 and 4 the analysis as done is not rigorous enough for the conclusions that authors are attempting to make. For example, in these tables the p-value listed is from a Kruskal-Wallis test (the authors should use this term rather than Krushkall-Wallis as listed throughout the manuscript). Thus, these p-values are from main effects, but it it not clear what the p-value is for the individual comparison between high, normal, low IL-1beta/IL-10 conditions and the non-ASD group. The authors describe changes between individual groups in the text, but the p-value for these direct comparison is not given. For examples on page 5 the authors state that the production of TNF and IL-10 were higher in the IL-1/IL-10 ratio ASD group as compared to the normal and low ratio ASDsubgroups -- no p value for these direct comparisons is given in the manuscript. Related to this, a major issue with the analyses performed is that the authors state that they use Mann Whitney tests (a type of t-test) for comparison of values between 2 groups. However, for each cytokine there are 4 conditions (high, normal, low, and non-ASD). Thus, a direct t-test is inappropriate and the authors should perform a one-way ANOVA with proper post-hoc to correct for multiple comparisons for each cytokine. Also, it is not clear why the authors are using non-parametric tests instead of parametric comparisons. The use of non-parametric tests would only be justified in the data fail to follow a normal distribution (which is not explicitly stated in the method section). Was this the case for all of the comparisons for each cytokine? If not, why weren't parametric stats used?
Author Response
The data presented in Tables 3 and 4 were re-analyzed with Kruskal-wallis one way ANOVA rank test. We used this method, since the data were not normally distributed. Also added the detailed results of differences of the 2 study groups as footnotes in Tables 3 and 4; differences are based on Mann-Whitney test. It is of note that cytokine production data in the absence of stimuli were added to Tables 3 and 4 in response to the reviewer #1' request.
Round 2
Reviewer 1 Report
NA
Author Response
Thank you for accepting the revised manuscript. You comments have been helpful.
Reviewer 2 Report
One of my major concerns with the first submission is that the authors are making direct comparisons between groups in several of the tables (e.g., table 3 and table 4) using the Mann-Whitney test, which does not correct the critical p value needed for multiple comparisons. For example, in these tables there are 4 groups. In order to make direct comparison between individual conditions in these groups the Mann-Whitney test is not appropriate as it does not correct for multiple comparisons and thus inflates the level of type 1 error. The authors need to perform a post-hoc following the ANOVA that takes corrects the p value for multiple comparisons. As it stands many of the direct comparison that have p values of less than 0.05 in this manuscript would likely not hold up with multiple comparison corrections. Thus, it is not clear how this would affect the interpretation of the results. It in not clear why the authors would report main effects using an Krushkall-wallis one way ANOVA then not use a post-hoc test following this ANOVA to assess direct comparisons. Without this appropriate use of statistics I am hesitant to support this manuscript.
Author Response
Thank you for your constructive comments. Your comments have been helpful for improving clarity of our data.
In the revised Tables 4 and 5, p values from one way ANOVA reports are shown. Results of ANOVA for comparison between groups are shown in the footnotes.
Normality of numerical data were assessed by skewness and kurtosis (Omnibus) with α=0.2. Assessment of normality was explained in the method section (Page 17, Lines 40-41). Except for CCL2 data under the culture of medium only, equal normality was rejected in all other culture conditions with this analysis.
Round 3
Reviewer 2 Report
The authors have properly addressed my concerns.